

**GEST: A multi-scale dynamics-based reconstruction of global ocean surface current**
**Guiyu Wang [1], Ge Chen [1], Chuanchuan Cao [1], Xiaoyan Chen [1], and Baoxiang Huang [2]**
[1]Frontiers Science Center for Deep Ocean Multispheres and Earth System, School of Marine
Technology, Ocean University of China, Qingdao, China
[2]Department of Computer Science and Technology, College of Computer Science and Technology,
Qingdao University, Qingdao, China
*Corresponding to*: G.Y. Wang (guiyuwang@stu.ouc.edu.cn)
**Abstract**   A high precision and fine resolution reconstruction of sea surface current is beneficial
to the exploration of complicated ocean dynamic processes. Existing studies mainly use satellite
sea level and wind stress fields to derive sea surface geostrophic and Ekman currents, and build
physical inversion models for global or regional oceans. Despite the obvious success, there are a
variety of typical dynamic processes in the ocean such as mesoscale eddies and small-scale
waves, and any product of surface current that neglects the contribution of wave motion would
be, at best, incomplete. In this context, we present a precise sea surface current product at 15 m
depth named GEST (Geostrophic-Ekman-Stokes-Tide) by analyzing the coupling relationship
between ocean surface components that correspond to different physical processes and the actual
currents as observed by drifting buoys. The GEST was generated based on Ekman, geostrophic
currents, and waved-induced Stokes drift and TPXO9 tidal currents. Specifically, in the
calculation of Ekman currents, local applicability is taken into account to ensure that the friction
layer of wind-driven current can reach where drifters could operate as normal. A comparison
proves that combining multi-scale theory and the condition of local applicability improves
estimation results by up to 3.6 cm/s compared with OSCAR product, and 0.3 cm/s with
GlobCurrent. Furthermore, by comparing the reconstructed products with 1° and 0.25°
resolution, we find that the higher resolution not only reveals more details of ocean currents
especially mesoscale eddy energy associated with geostrophic currents, but also improves the
accuracy by up to 5.62 cm/s.



## 1 Introduction

The ocean surface current is fundamental for climate studies and is closely associated with many oceanographic applications (Lagerloef et al., 2003; Caniaux et al., 2005; Jin et al., 2014). Detailed information of these currents significantly affects fisheries management, shipping control and ocean rescue (Lo et al., 1998). In particular, the knowledge of velocities and directions of ocean surface current is necessary for the tracking of suspended matter (such as spilled oil, algae, sea ice, and floating microplastic) (Choi et al., 2013; Yang et al., 2014; Weisberg et al., 2019; Onink et al., 2019), which can effectively explore plankton ecology and safeguard marine ecology (Wahl et al., 1996; Garçon et al., 2001). Therefore, high-resolution, long time series of ocean surface current are needed to extend existing oceanographic datasets, deepen the understanding of refined ocean dynamic processes, and make accurate predictions about changes in the climate system (Chapman et al., 2017).

Emerging technologies have been effectively applied to the observation of ocean surface current, but there is no global current observation system to obtain sea surface velocities on a full spatial and temporal scale so far. Despite the obvious success of the Global Drifter Program (GDP) issued by the National Oceanic and Atmospheric Administration (NOAA) which has run for about 40 years, limitations and shortcomings like coarse space-time sampling and few coastal distributions are also existing.

It is well understood that satellite-based algorithms can explain about 70 % of the global ocean current dynamics (Sudre and Morrow, 2008), in which large-scale geostrophic circulation has been determined over the last 25-30 years. When forming a steady balance between wind-induced frictional stresses and Coriolis forces, Ekman current is the theoretical basis for understanding wind-driven ocean component, and is similarly considered as another main dynamic mechanism of ocean currents (Ekman, 1905). Remarkable progress has been made in estimating long time series ocean surface current datasets following the construction of global physical estimation model, such as the OSCAR (Ocean Surface Current Analysis Real Time), GEKCO (Geostrophic and Ekman Current Observatory) and the GlobCurrent project (Bonjean and Lagerloef, 2002; Dohan and Maximenko, 2010; Sudre et al., 2013; Rio et al., 2014). Sudre et al. (2013) obtained wind-driven currents by reanalyzing wind stress fields based on a two-parameter model, in which parameters were modified by linear stability equations based on the assimilation of drifter data.



The global geostrophic component was determined from the gradients ($\partial h/\partial y$, $\partial h/\partial x$) of the ocean
sea surface height (SSH), which was calculated from the sum of the altimeter sea-level anomaly
(SLA) and the mean dynamic topography (MDT). In the equator, a β-plane geostrophic
approximation proposed by Lagerloef et al. (1999) was adopted because of the vanishing of the
Coriolis force. Rio et al. (2014) used a new 0.25° CNES-CLS13 MDT by assimilating drifter
observations, with which the velocities in strong currents were increased by 200 % on average. In
addition, they also calculated a two-level (0 m and 15 m) empirical Ekman model with surface
drifters and Argo floats, and both piece of algorithms lead to more accurate ocean currents.
Bonjean and Lagerloef (2002) gained the improved estimate of tropical circulation with
geostrophic, Ekman, Stommel shear dynamics, and a complementary term from surface buoyancy
gradient, with quasi-linear and steady physics. All these surface currents were directly calculated
from geostrophic and wind-induced Ekman currents. In the actual ocean, however, the movement
of upper oceans is the result of multiple environmental driving mechanisms, and can be broadly
divided into large-scale ocean circulations, micro scale internal waves and storm surges. Wave-
induced transport acts either as an enhancement or attenuation to the mean flow within Ekman
layer, and should also be regarded as an essential source of the ocean current (Bi et al., 2012).
In this paper, we examine the inherent coupling relationship between velocities measured by
drifters and components of ocean surface current, including geostrophic and Ekman currents,
wave-induced tidal currents and Stokes drift, which results in a more accurate ocean surface
current product incorporating multi-scale dynamic mechanisms. This global daily product covers
the period of 2013-2019, with a 0.25° spatial resolution, and is compared with the OSCAR product
and the GlobCurrent project.
The organization of the remaining parts of this paper is as follows. Section 2 is concerned with the
data used to calculate the components of the flow field. Section 3 presents the method to
reconstruct ocean surface current. Section 4 shows the results of flow field estimates and provides
a discussion of our estimation work. Finally, we reveal the conclusions in Sect. 5.
**2 Data**
2.1 Tidal data
Recently, many tidal estimation models with different resolutions and precision have emerged



because of insufficient tidal observation stations (e.g., HAMTIDE, FES, and TPXO) (Lyard et al.,
2006; Taguchi et al., 2014). We use the tidal data from the latest version of TPXO models available
at the time of the study, the TPXO9 model, which is based on the two-dimensional positive
pressure fluid equation, applying the generalized inversion method to assimilate altimetry data
(e.g., satellite altimetry from T/P, ERS1, 2, Envisat satellites) and in situ station-based tidal data,
and then fitting the data with the least squares (LS) (Egbert et al., 2002; Egbert et al., 2018).
Established by Gary Egbert and Lanna Erofeeva from Oregon State University, USA, the TPXO9-
atlas we used for the global hourly tidal data set has accurate and reliable results, which is reflected
in the low root mean square error of M2 constituents in the deep ocean, the shallow sea, and the
continental shelf (Sun et al., 2022). The derived tidal data has a 1/6° resolution and contains the
harmonic constant of 15 constituents, including eight major constituents (M2, S2, N2, K2, K1, O1,
P1, and Q1).
2.2 Wind data
The global daily wind fields we used were retrieved from QuikSCAT and Windsat satellites
launched by the United States during in-orbit operation, with a cell size of $0.25° \times 0.25°$, spanning
14 years from January 2006 to January 2019. This data sets were then used to calculate wind stress
fields by a bulk formula, which refers to a drag coefficient ($C_d$) that curve-fits data for low-to-
moderate winds with data for high wind speeds (Oey et al., 2006).
Based on the reanalysis wind stress fields, we aim to obtain wind-induced Ekman currents, which
are perpendicular and shifted to the left of the direction of wind stress in the southern hemisphere
(the opposite in the Northern Hemisphere), forming a spiral in response to the turbulent viscosity
that depends on the local ocean state (Roach et al., 2015).
2.3 CMEMS global geostrophic currents and Stokes drift
The reanalysis daily geostrophic velocities generated by Copernicus Marine Environment
Monitoring Service (CMEMS) with a grid size of $(0.25)° \times (0.25)°$ are used in this study (Pujol et
al., 2016) covering the years 2006-2019. For latitudes outside ±5° N, geostrophic currents are
obtained from altimeter maps of the Absolute Dynamic Topography (ADT) using a nine-point
stencil width methodology. While near the equatorial zone, the geostrophic balance is not
applicable anymore and a β-plane approximation of Lagerloef et al. (1999) is introduced.



In addition, the Stokes drift is taken from the Global Ocean Waves Reanalysis (WAVERYS)
numerical assimilation products of CMEMS at grid points with 0.2° longitude and 0.2° latitude
spacing covering the years 2006 to 2019. These products involve 3-hour instantaneous fields of
integrated wave parameters (e.g., Sea surface wave significant height (SWH), sea surface wind
wave from direction (WW), and Stokes drift velocities (VSDXY)), and are derived from the global
wave reanalysis of the MeteoFranceWAve-Model (MFWAM) wave model by assimilating the
altimeter wave data and directional spectrum provided by Sentinel-1 (Law-Chune et al., 2021).

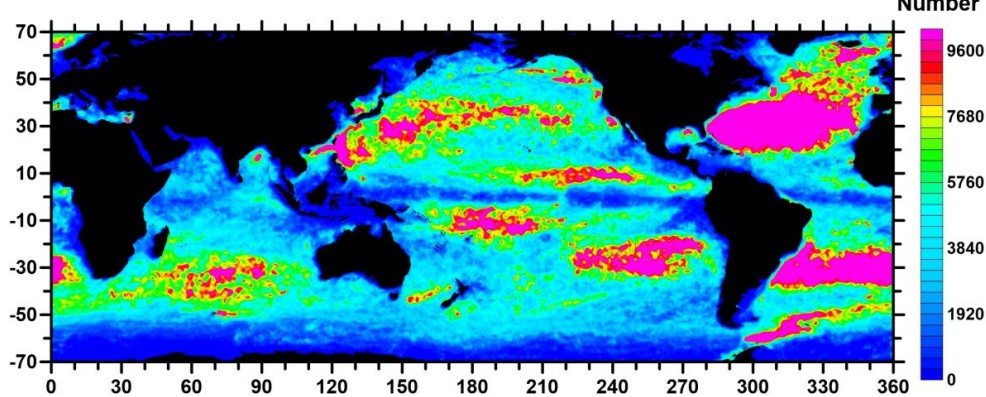

**Figure 1:** Spatial distribution for the drifter observations per $1° \times 1°$ from 1999 to 2019.
2.4 Surface drifting buoys
AOML's Global Drift Program consists of a global array of more than 1,000 satellite-tracked
surface drifting buoys ("drifters") that take large scale measurements of surface velocities and
track the direction of currents by Lagrangian method. In this study, we use the velocities of these
in situ drifters as 'reference tags'. This global data set with long time series is interpolated at hourly
intervals and contains rich details of global high-frequency and small-scale ocean processes.
Positions and velocities are derived by locally modelling the trajectories to construct first-order
polynomials, whose coefficients are obtained by maximizing the likelihood function (Elipot et al.,
2016; Elipot et al., 2022). To give an idea about the collocated drifter data set, the global spatial
distribution of the drifter observations per $1°\times1°$ cell is shown in Fig. 1.
**3 Reconstruction of sea surface current at 15 m depth**



3.1 Local applicability of wind-driven currents
We use the two-parameter empirical model displayed in Eq. (1) to estimate Ekman Currents $(\boldsymbol{u_e}, \boldsymbol{v_e})$
(Niiler and Paduan, 1995; Sudre and Morrow, 2008),
$$(\boldsymbol{u_e} + i\boldsymbol{v_e}) = Be^{i\theta}\left(\boldsymbol{\tau_x} + i\boldsymbol{\tau_y}\right), \tag{1}$$
where $\tau_x$ and $\tau_y$ represent zonal and meridional wind stress. In the region out of 25° S-25° N, we
follow the amplitude $B = 0.3$ m s$^{-1}$ pa$^{-1}$ and the vectoring angle $\theta = 55°$ with respect to the wind
stress direction. Near the equator (25° S-25° N), the latitude-dependent parameters $B$ and $\theta$ derived
by Lagerloef et al. (1999) are adopted which are related to the Coriolis force parameter $f$, the water
density $\rho = 1025$ kg m$^{-3}$, the linear drag coefficient $r_e = 2.15 \times 10^{-4}$ m s$^{-1}$, and the friction depth
($h_e$=32.5 m),
$$B = \frac{1}{\rho}\left(r_e{}^2 + f^2 h_e{}^2\right)^{-\frac{1}{2}} \tag{2a}$$
$$\theta = \arctan\left(\frac{fh_e}{r_e}\right). \tag{2b}$$
A climatological annual mean of Ekman currents as well as the geostrophic, Stokes, and tidal
components is shown in Fig. 2. It seems that the uncertain assessment scheme of parameters ($B$
and $\theta$) within equatorial band proposed by Lagerloef et al. (1999) leads to high Ekman current
velocities, which contradicts the theory of calm belt near the equator. Although previous studies
have calibrated parameters ($B$, $\theta$) by using the least squares fit based on ageostrophic wind-driven
velocities obtained by subtracting geostrophic component from the drifting buoy velocities, it does
not constitute a fully independent validation with our reference field in this study. Accurate and
independent parameter scheme of wind-driven velocities may be an improvement of this work in
the future.
Subject to wind stress intensity and the viscosity coefficient of seawater, the friction depth of wind-
driven currents (or Ekman depth) is locally adaptive, ranging from a few meters to hundred meters.
A verification is necessary that the mixing depth of the Ekman layer reaches the position of the
drogued drifters. Following a linear steady momentum balance in Eq. (3a) and Eq. (3b) proposed

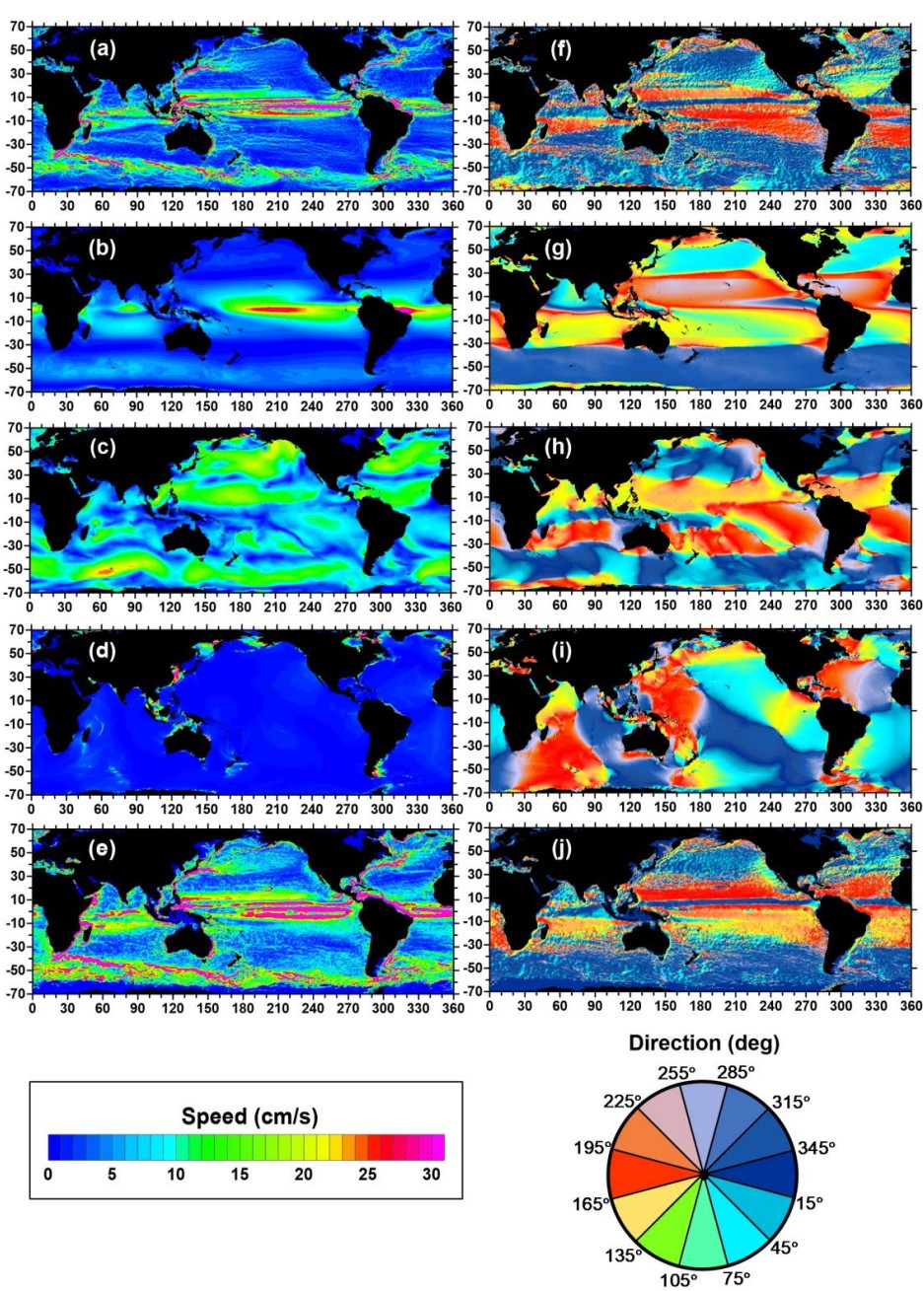

**Figure 2:** Global annual climatological flow fields. The left column represents flow speed of (a) geostrophic currents for 1999-2019, (b) Ekman currents for 1999-2019, (c) Stokes drift for 2013-2019, (d) tidal currents for 2013-2019, and (e) drifter observations for 1999-2019. While the right column is the flow direction.

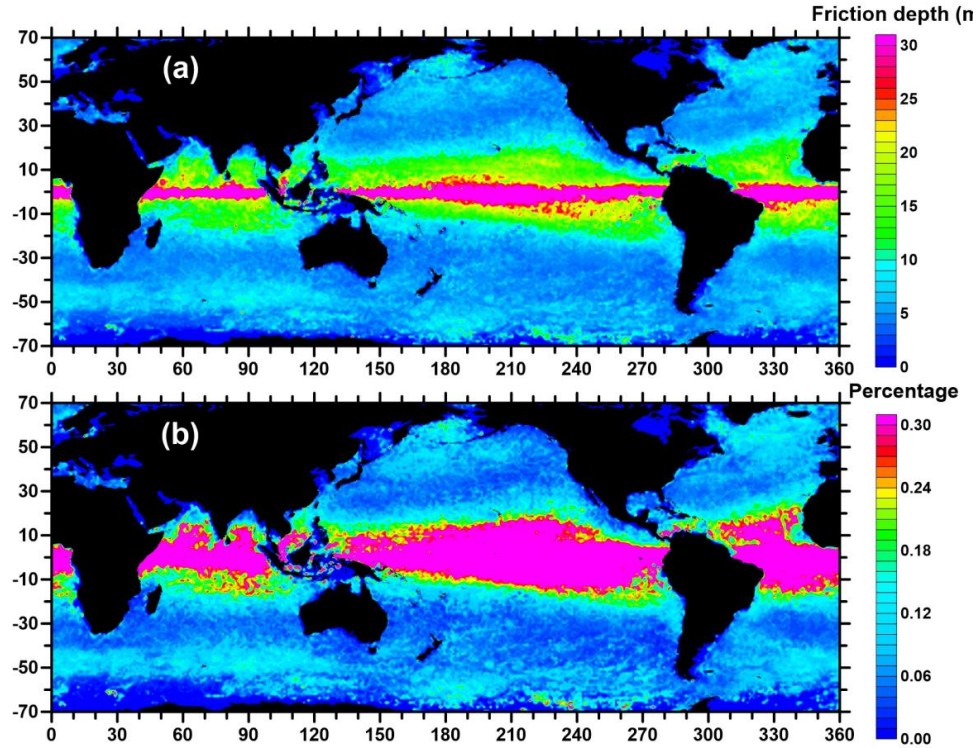

**Figure 3:** (a) Global mean distribution of the friction depth per 1° × 1°. (b) Proportion of friction

depth up to 15 m in drifter observations.

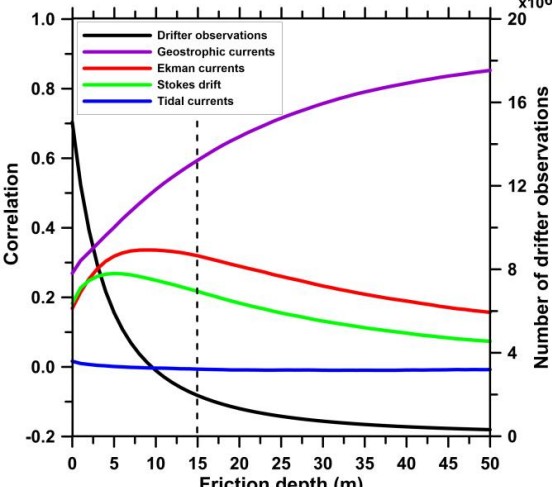

**Figure 4:** The correlation between four flow fields and observed velocity of drifters (left axis),

and the number of drifter observations varying with Ekman depth $h_e$ (right axis).



by Meurs and Niiler (1997) and Lagerloef et al. (1999), the friction depth $h_e$ of Ekman currents
can be expressed by Eq. (3c),

$$f h_e \boldsymbol{u} + r_e \boldsymbol{v} = \boldsymbol{\tau_y}/\rho \tag{3a}$$

$$r_e \boldsymbol{u} - f h_e \boldsymbol{v} = \boldsymbol{\tau_x}/\rho \tag{3b}$$

$$h_e = \frac{1}{f} \frac{\boldsymbol{\tau_y} u - \boldsymbol{\tau_x} v}{\rho \, (u^2 + v^2)}, \tag{3c}$$

where $u$ and $v$ are ageostrophic vectors when removing geostrophic velocities from drifters
observations of monthly climate state. Figure 3a illustrates a global climatological distribution of
Ekman depth per 1° grid. With the deepening of Ekman depth $h_e$, the correlation between currents
components and drifter velocities shows a trend of increase and then decrease, with the latter
possibly related to the sharp decline in the number of drifter observations after the validation of
wind-driven friction depth (Fig. 4). We select 15 m friction depth as the minimum threshold for

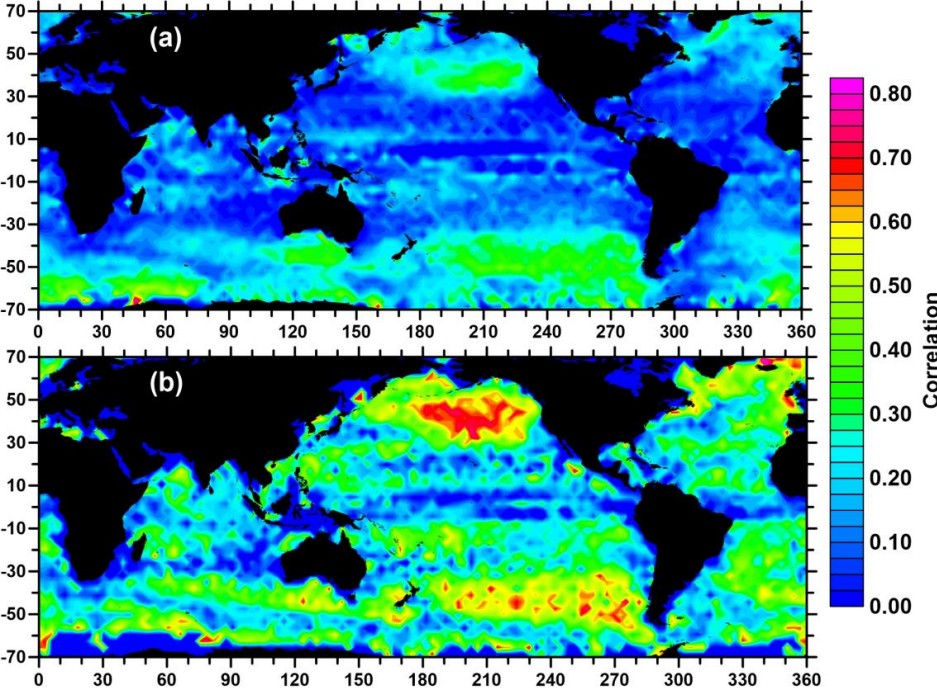

**Figure 5:** The correlation between Ekman currents and drifter observations per 3° grid (a) before
and (b) after depth validation.





the filtration of Ekman currents datasets by comprehensively considering the location of drogued
drifters, the correlation between ocean components and drifter observations, and the change in the
number of drifter observations. The ratio of drifter observations filtered by the selected friction
depth (15 m) to all drifter observations indicates that about 9/10 of the quantities have been culled
(Fig. 3b). Figures 5a and 5b show that the correlation between Ekman currents and drifter
observations increases by ~15 % after the filtration.

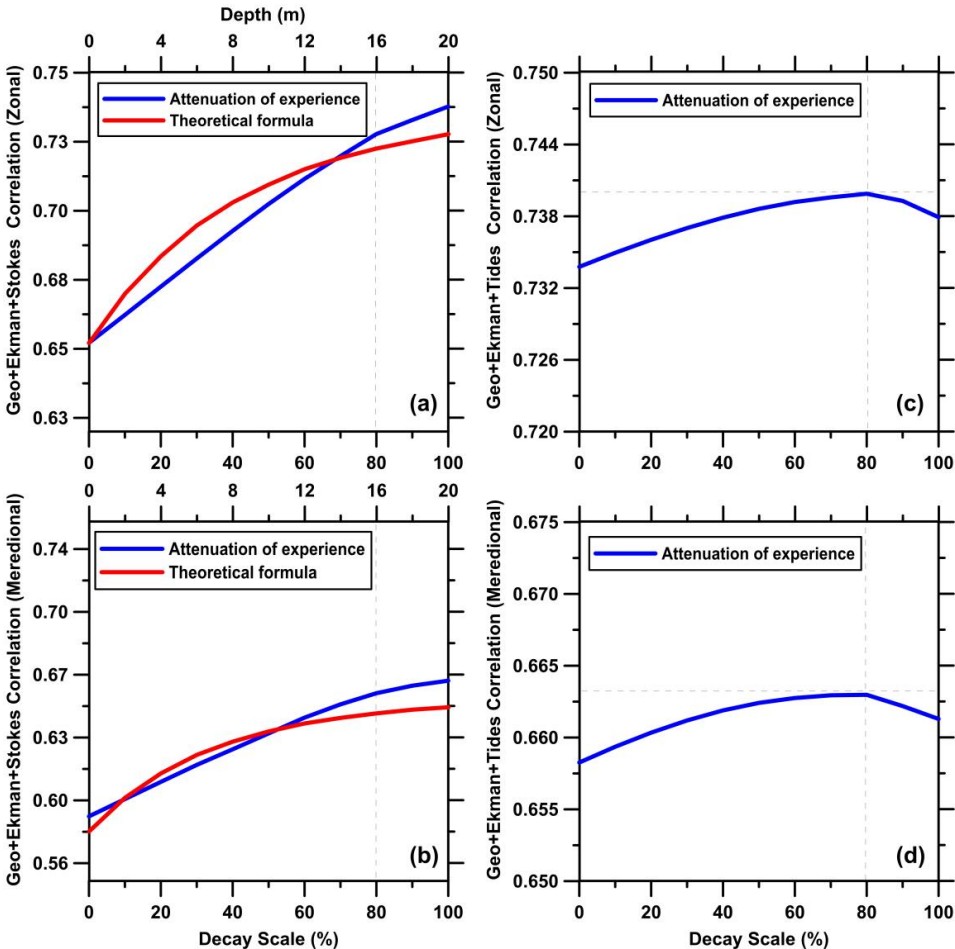


**Figure 6:** The correlation of the zonal (top) and meridional (bottom) vector combinations of (a)-
(b) geostrophic, Ekman currents, and Stokes drift, decaying with depth (top coordinate) and
percentage (bottom coordinate), and (c)-(d) geostrophic, Ekman currents, and tidal currents
decaying with percentage.



3.2 Decay scales of Stokes drift and tidal currents

The reanalysis Stokes drift and tidal currents cover 0 m vertically and need to be attenuated by empirical percentage method and exponential method that can be expressed in Eq. (4),

$$\boldsymbol{u}_s = \boldsymbol{u}_{s0} \exp(2kz), \tag{4}$$

where $u_{s0}$ denotes the surface Stokes drift, $z$ is the profile derived from a monochromatic wave with wavenumber $k$ and wavelength (Kukulka and Harcourt, 2017). As shown in Figs. 6a-6d, the correlation between drifter observations and different vector combinations of geostrophic, Ekman, tidal currents, and Stokes drift, increases ceaselessly with depth when using the theoretical formula and, for tides, reaches peaks at 80 % attenuation when adopting the percentage decay method. Thus, the 80 % decay ratio is used for Stokes drift and tidal currents before the subsequent estimation of surface current.

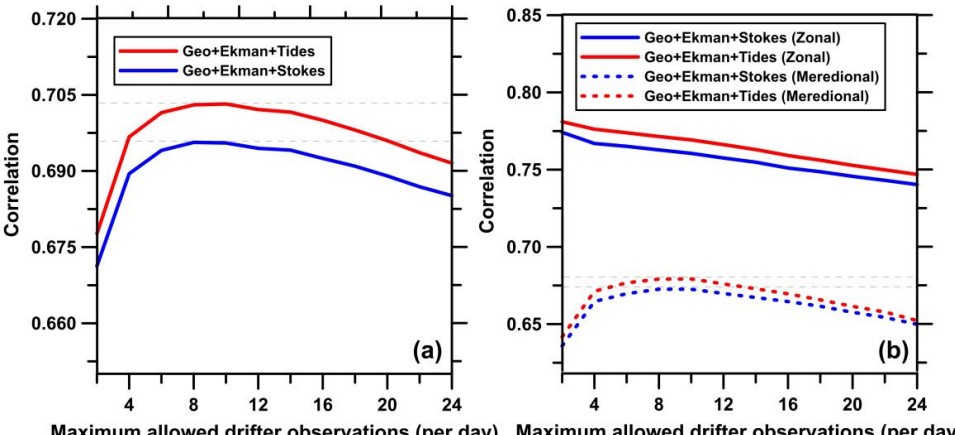

**Figure 7:** The correlation of different combinations of geostrophic, Ekman, tidal currents, and Stokes drift varying with maximum allowed drifter observations per day corresponding to (a) vector synthesis and (b) zonal and meridional vector.

3.3 Optimal time matching interval between wind fields and drifter observations

Exact correspondence matching with the retrieved time of wind field will result in insufficient drift




observations. Based on a series of experiments, we confirm that the correlation between
synthesized vectors or their components of geostrophic, Ekman, tidal currents, and Stokes drift
(Figs. 7a and 7b) and the maximum allowed drifter observations per day peaks near the number of
12 (i.e., the time interval is no more than 6 hours).
3.4 Correlation analysis of ocean current components with drifter observations
A new drifter data set has been constructed by matching the pre-processed geostrophic currents,
Ekman currents, Stokes drift and tidal currents. We then reveal the coefficient of the Pearson
correlation between ocean components and drifter observations referring to Eq. (5) which is related
to true observations $(x_i, y_i)$ and the averaged ones $(\overline{x}, \overline{y})$,

$$Pearson = \frac{\sum_{i-1}^{n} (x_i - \overline{x}) (y_i - \overline{y})}{\sqrt{\sum_{i=1}^{n} (x_i - \overline{x})^2} \sqrt{\sum_{i=1}^{n} (y_i - \overline{y})^2}} \; . \tag{5}$$

Going through Figs. 8a-8d, the contribution of each component to the flow field is clearly evident.
Geostrophic currents act as the primary mechanism that form the ocean surface current field, and
the Pearson correlation coefficient can reach nearly 0.98 in the regions with strong and persistent

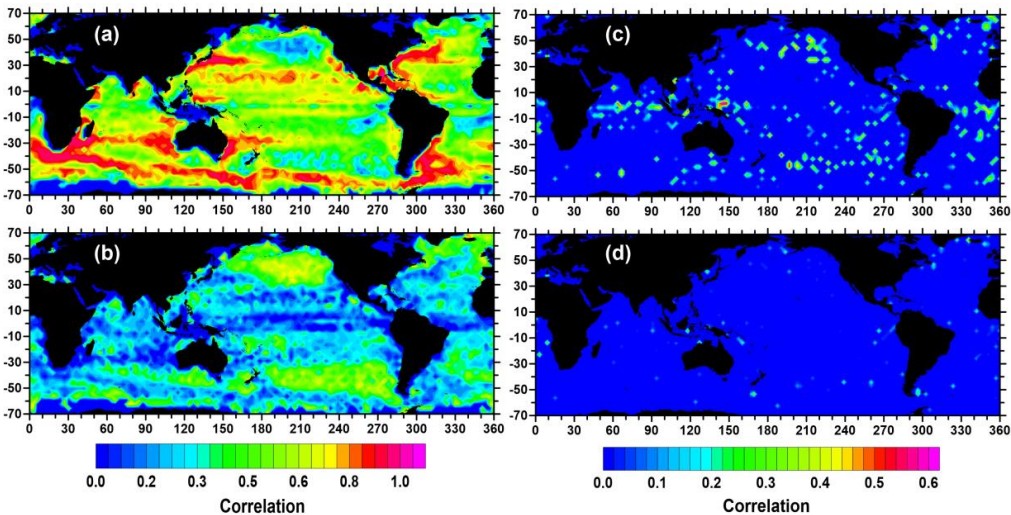

**Figure 8:** Geographical distribution of the Pearson correlation coefficient of (a) geostrophic currents, (b) Ekman currents, (c) Stokes drift, and (d) tidal currents.



currents along the western boundary. Moreover, wind-induced Ekman currents contribute
significantly in the westerly zone of the northern and southern hemispheres, while are not
noticeable near the equator, which may be related to the calm zone in the tropics. At the same time,
Stokes drift can contribute up to 60 % in the West Pacific, while tidal currents work mainly in the
shallow coastal area with the maximum value of 0.5, which is consistent with its higher velocities
along the coast.
3.5 Optimal GEST model for estimating ocean surface current
Given the seasonal and local variability in velocity and direction of the surface current, we first
build least-squares linear regression (Eq. (6)) on a 3° grid by season using different flow field
combinations (Sub-GE (Geostrophic-Ekman), Sub-GES (Geostrophic-Ekman-Stokes), Sub-GET
(Geostrophic-Ekman-Tide), and Sub-GEST (Geostrophic-Ekman-Stokes-Tide)) to determine the
optimal configuration of the flow field components,

$$E = \sum_{i=0}^{n} e_i{}^2 = \sum_{i=1}^{n} (y_i - \hat{y}_i)^2 , \qquad (6)$$

where $\hat{y}_i$ is the predicted observation, $y_i$ is the true ones, and $n$ is the number of observations. Four
initial datasets, called GEST-1, were obtained based on the sub-models above (Sub-GE, Sub-GES,
Sub-GET, Sub-GEST) by using the data from 2006-2012. We verified the error of the GEST-1
versus drifter data with root mean square error (RMSE),

$$RMSE = \sqrt{\frac{1}{n} \sum_{n=1}^{n} (\hat{y}_i - y_i)^2} . \qquad (7)$$

Table 1 shows a regional comparison regarding the reconstruction errors of the sub-models. In the
Antarctic Circumpolar Current (ACC) where the GE model has a higher accuracy, it seems that
the inclusion of tidal currents and Stokes drift contributes to signal noise, while in the equatorial
regions of Pacific and Indian Ocean, the results of the Sub-GEST are significantly better than other
combinations. In addition, compared with Sub-GE model (Figs. 9a and 9b), the Sub-GES is more
effective mainly in the western equatorial Pacific, such as the Peninsular Malaysia and Marshall
Islands (Figs. 9e and 9f). Furthermore, as shown in Figs. 9g and 9h, the averaged RMSE of Sub-



GET in the south coast of Kyushu Island decreased to 11.35 cm/s from about 11.75 cm/s of the
Sub-GE (Fig. 9c), and the Gulf of Mexico also decreased about 0.3 cm/s than Sub-GE (Fig. 9d)
with a maximum decrease of up to 10 % in this region, which is likely related to frequent tropical

**Table 1.** Verified RMSE (cm/s) based on Sub-GE/ Sub-GES/ Sub-GET/ Sub-GEST models

| Reconstruction Model | Longitude and Latitude | | | |
| --- | --- | --- | --- | --- |
| | 55° E-70° E 10° S-5° N | 80° E-100° E 45° S-60° S | 103° E-113° E 3° N-19° N | 125° E-142° E 22° N-37° N |
| Sub-GE | 22.0333 | 9.1091 | 12.3418 | 11.7525 |
| Sub-GES | 20.9319 | 9.2731 | 11.9773 | 11.9136 |
| Sub-GET | 23.3724 | 9.2368 | 12.4657 | 11.3579 |
| Sub-GEST | 20.6216 | 9.3872 | 12.0061 | 11.4845 |

| Reconstruction Model | Longitude and Latitude | | | |
| --- | --- | --- | --- | --- |
| | 156° E-173° E 10° S-10° N | 175° E-195° E 40° S-55° S | 228° E-240° E 9° S-21° S | 260° E-290° E 18° N-28° N |
| Sub-GE | 16.9963 | 7.1911 | 6.5527 | 7.9553 |
| Sub-GES | 16.4886 | 7.3135 | 6.5541 | 8.0693 |
| Sub-GET | 17.0196 | 7.2609 | 6.5225 | 7.6361 |
| Sub-GEST | 16.5005 | 7.3945 | 6.5184 | 7.8609 |

cyclone activities in the Gulf of Mexico and the southern coast of Kyushu Island. From August to
October, the South Kyushu Island is vulnerable to tropical cyclones in the Northwest Pacific Ocean,
while storm surges in the north dominate in the Gulf of Mexico. Driven by typhoons, seawater





rises steeply to form storm surges, with water levels sometimes reaching 5 m. When storm surges
are superimposed on astronomical surges, the magnitude of water level changes could be even
greater (Murty et al., 1986; Dube et al., 1997; Bilskie et al., 2016).

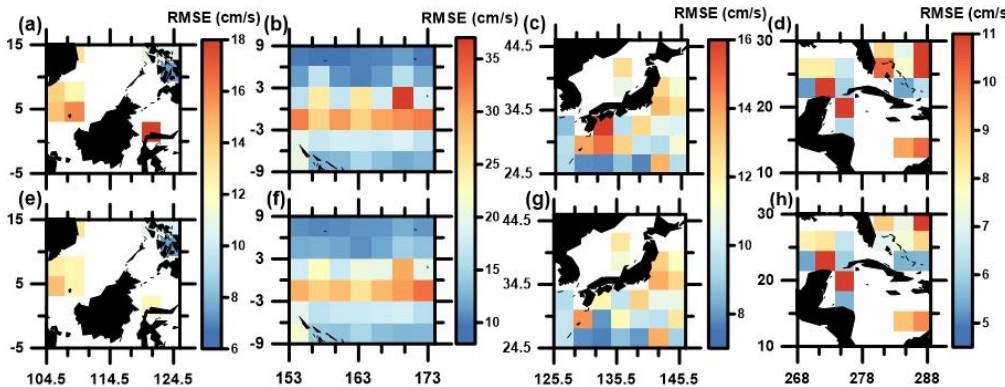

**Figure 9:** RMSE of (a)-(d) Sub-GE, (e)-(f) Sub-GES, and (g)-(h) Sub-GET models.
As mentioned above, wave-induced Stokes drift and tidal currents contribute significantly only in
certain regions (see Table 1) and should not be joined entirely on a global scale. Thus, the optimal
GEST model was chosen from the above sub-models (Sub-GE, Sub-GES, Sub-GET, and Sub-
GEST) per $3° \times 3°$ grid by the error validation of the regression, and then was used for the estimates
of ocean surface current incorporating geostrophic and Ekman currents, Stokes drift, and tidal
currents.
**4 Result and discussion**
4.1 GEST products and associated verification
Following the optimized reconstruction scheme outlined above, a 0.25° resolution data set named
GEST, such as the daily current velocity on a particular reference date (Fig. 10), was obtained
combining the geostrophic flow with other field components. We have also considered building
products with 1° resolution by matching other flow fields with Ekman currents, but quantitative
validations showed that the 0.25° reconstructed field significantly decreased global averaged
RMSE from about 14.61 cm/s to 9.36 cm/s over the 1° resolution product. Figures. 11a-11c show

Earth System
Science
Data

the global distribution of the RMSE for 1° product, with significant errors in the western boundary
currents and the Kuroshio area. It appears that intense kinetic energies of mesoscale eddies exist
in these regions (Fig. 12) and likely neglected by the coarse spatial resolution (Chen and Han,

283    2019).

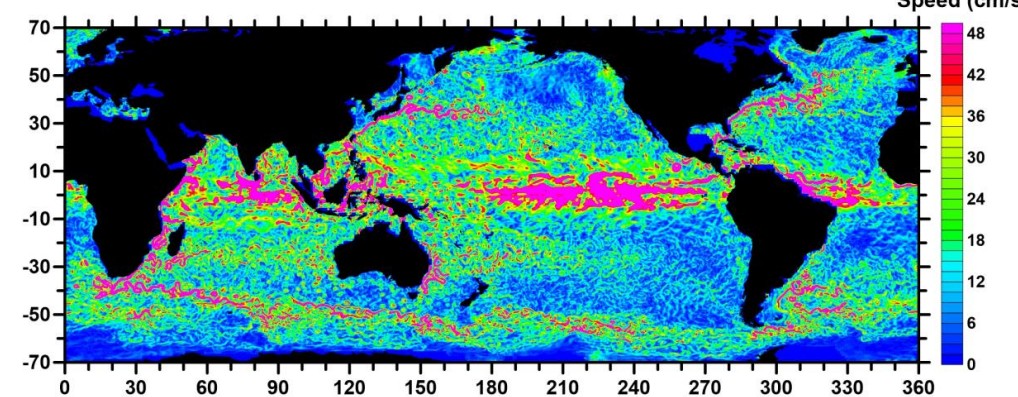


**Figure 10:** Daily current field of the GEST product on 01 January 2017.

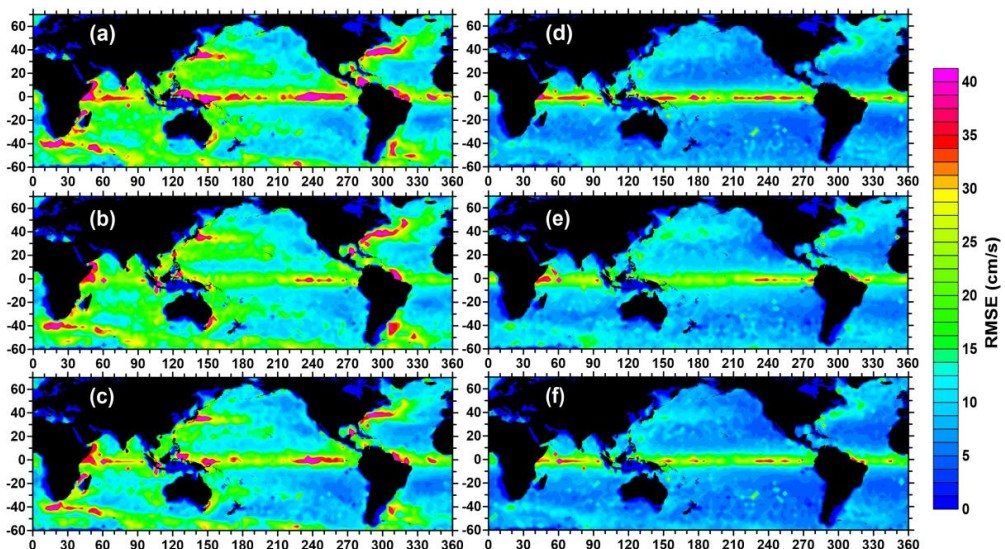


**Figure 11:** The RMSE of the 1° reconstructed field of (a) zonal vector, (b) meridional vector,
and (c) synthetic vector, and the 0.25° reconstructed field of (d) zonal vector, (e) meridional
vector, and (f) synthetic vector with the optimal GEST model.





Moreover, compared with the estimates of regression theory, we find that the optimal vector
synthesis scheme has a higher reconstruction accuracy in estimating the 0.25° resolution field, and
additionally, traditional geostrophic and Ekman currents model is also available in regions without
reference velocities. As shown in Figs. 11d-11f, the RMSE of the optimal vector synthesis method
(8.99 cm/s) decreases by almost 0.4 cm/s compared with the optimal regression method (9.36 cm/s).

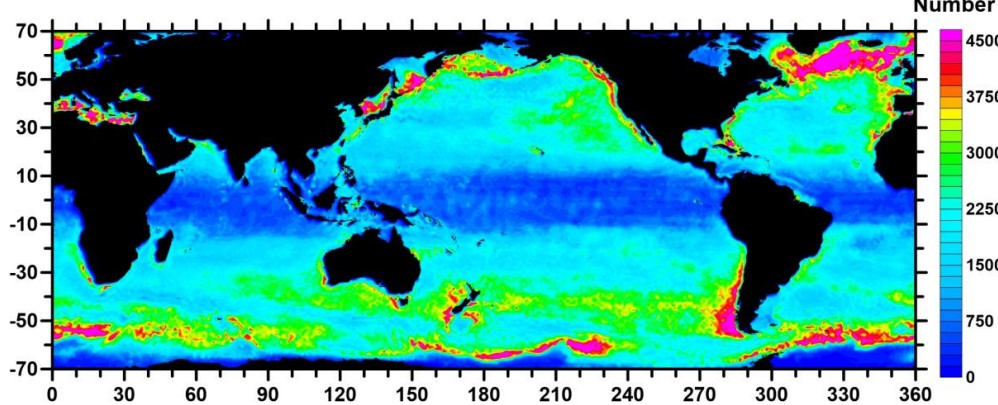


**Figure 12:** The global distribution of mesoscale eddies.
4.2 Comparison of the GEST to other global surface current products
Excellent achievements have been made in physical model-based reconstructions of ocean surface
current. The OSCAR near-surface current with a grid size of 1° on a 5 days basis use the quasi-
linear, quasi-steady sea surface momentum equations and improve the equatorial algorithm by
fitting 12 orthogonal polynomials (Johnson et al., 2007). Meanwhile, the GlobCurrent data set
presents an improved empirical Ekman model and a global 0.25° MDT by exploiting information
from geostrophic currents and the drogued drifters. Figures. 13a and 13c show the estimate errors
of the OSCAR and GlobCurrent products when compared with the velocity of the drogued drifters.
The RMSE of GlobCurrent is, on average, ~2 cm/s smaller than OSCAR data due to the coarse
temporal resolution and finite spatial distribution of T/P and Jason series altimeter data used in
OSCAR currents. Another promising finding is that a reduction of up to 7-8 cm/s in RMSE is
observed for both products when compared to drifters of which friction depths of matched Ekman
currents are up to 15 m (Figs. 13b and 13d).
In this paper, we are not able to further consider the drogue lost information because 90 % of



drifter observations are discarded with the condition of friction layer. As evidenced in Fig. 11f, the
RMSE of GEST product is reduced to 8.99 cm/s compared to 12.55 cm/s for the OSCAR product,
and 9.28 cm/s for the GlobCurrent, and especially in the eastern equatorial Pacific, our approach
is quite efficient with an error reduction of up to 10 cm/s than that of OSCAR and GlobCurrent.
In the tropical Indian Ocean and western Pacific regions, while the GEST currents show better
accuracy than the GlobCurrent, the OSCAR currents outperformed both products, possibly related
to the improved equatorial algorithm proposed by Bonjean and Lagerloef (2002).

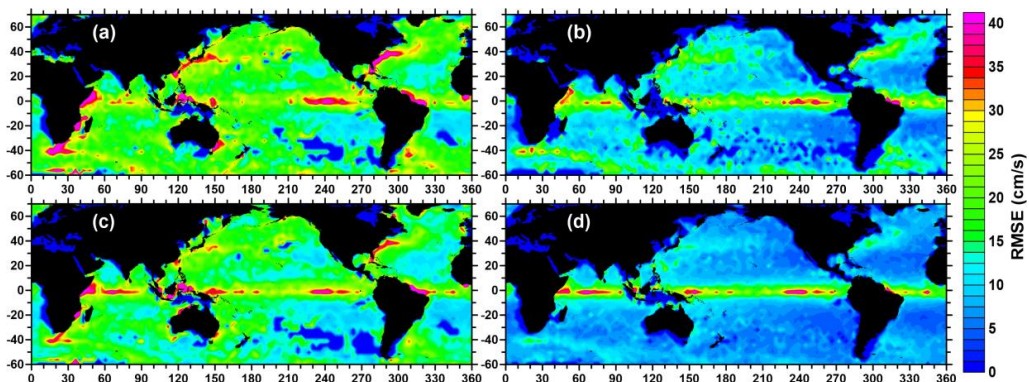


**Figure 13:** RMSE between (a) OSCAR currents and drogued drifters, (b) OSCAR currents and
the drifters verified by a friction depth of 15 m, (c) GlobCurrent and drogued drifters, (d)
GlobCurrent and the drifters verified by a friction depth of 15 m covering the years 2013-2017.
In addition, the reconstruction error with 1° latitudinal band was calculated for each product, as
well as the comparison of flow direction by radar maps, to assess the regional difference of the
estimation accuracy. It is quite evident that both GlobCurrent and GEST reflect better estimation
trends outside the equator due to the fine resolution (Fig. 14a). In particular, the GEST product
outperforms GlobCurrent within 60° S-50° N, which is consistent with research showing that tides
and Stokes drift reveal significant results in coastal and equatorial regions concentrating at low-
and mid-latitudes. At the equator, despite the lower spatial and temporal resolution of the OSCAR
data set, it also shows significant consequence with the drifter observations. Figure 14b shows that
GEST product has a better estimation of drifter velocity within 0°-270° angle range of the flow
direction than others. The lower efficiency at high latitudes and in the 270°-320° angle range is



likely related to Rio et al. (2014) assimilating the drift observations used as reference fields in this
paper to provide missing short-scale information in the calculation of the CNES-CLS13 MDT and
wind-driven parameters with spatio-temporal variability.

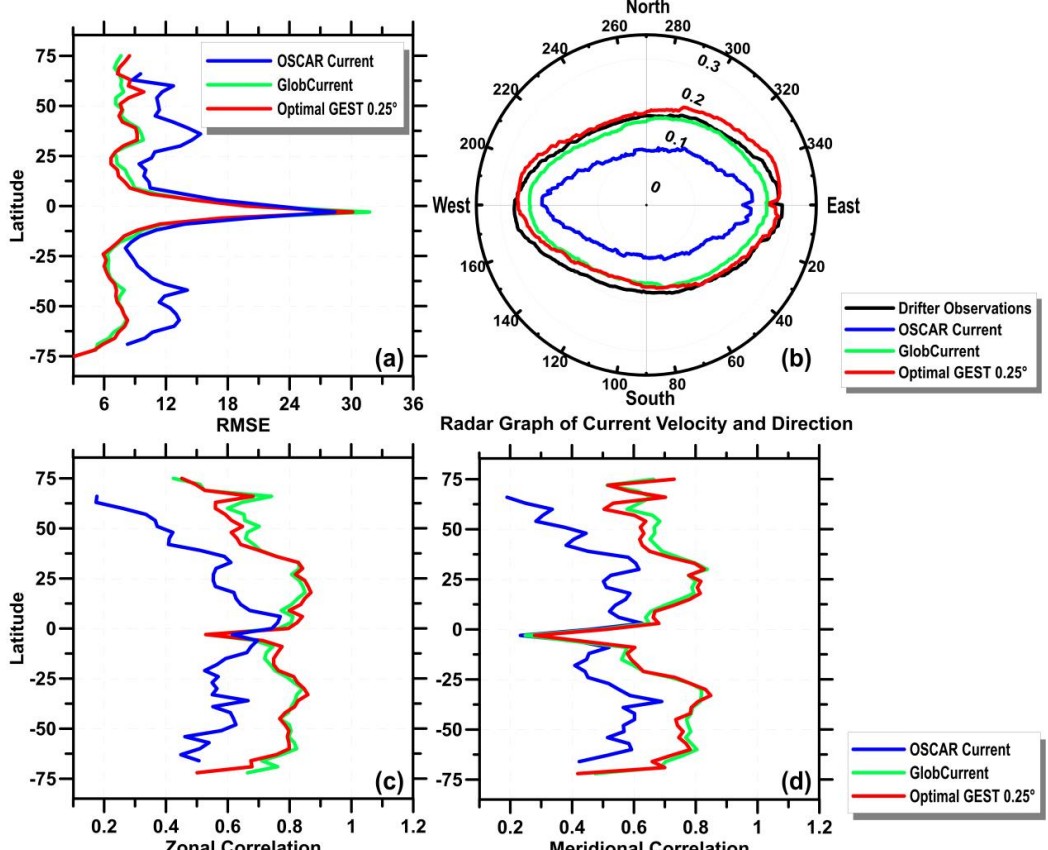


**Figure 14:** (a) RMSE based on latitudinal bands between each current product (the global
OSCAR current (blue line), the GlobCurrent (green line), the modified 0.25° GEST current (red
line)) and the surface drifters verified by a friction depth of 15 m. (b) A radar diagram of flow
velocity and direction versus the drifter observations (black line), the global OSCAR current
(blue line), the GlobCurrent (green line), and the modified GEST current (red line). (c) The zonal
correlations and (d) the meridional correlations for the global OSCAR current (blue line), the
GlobCurrent (green line), and the modified 0.25° GEST current (red line).



To further verify the robustness of the results, the correlation between the ocean current products
and drifter velocities in each latitudinal zone was calculated separately. As shown in Figs. 14c and
14d, the GEST product is behind GlobCurrent by about 5 % of the correlation within 40° and 55°
in the Northern Hemisphere. And it, in contrast, shows higher correlation coefficients at low to
mid latitudes of the Southern Hemisphere. In the equatorial region, both pieces of products
remained inferior to the OSCAR data for the horizontal component, being consistent with the
results of the RMSE. However, the GEST product delivers better results of meridional correlations
possibly due to the addition of Stokes drift.
It is notable that existing current products have poor estimates in the equatorial within 10° S-10°
N and western boundary currents. A plausible interpretation is that the parameters of wind-driven
model and the higher-order balance of geostrophic velocities are still uncertain. Furthermore,
previous studies have indicated that ocean eddies and large-scale planetary waves (e.g., Rossby
and Kelvin waves) correspond to ocean circulation at different scales. Especially in the equatorial
band, the fluctuation significantly contributes well to the upper ocean circulation. In particular, the
scale of the zonal component of the eastward Kelvin wave is more significant than that of the
meridional component (Forbes, 2000), which may influence the inversion accuracy of the zonal
feature in the equatorial region. Meanwhile, the complex aliasing effect between intense ocean
currents, eddies with strong kinetic energies, and large-scale planetary waves with strong
fluctuations, is also likely to be an important factor affecting the reconstruction accuracy in this
study (Hansen and Paul, 1984; Klocker and Marshall, 2014; Huang et al., 2021; Chen et al., 2022).
**5 Data availability**
All data used in this study are publicly available, and are listed in Table 2. The TPXO9-atlas for
tidal currents is available for academic research and other non-commercial uses under previous
registration from https://www.tpxo.net/global/tpxo9-atlas (Egbert et al., 2018). The GEST
product produced in this research can be found at https://doi.org/10.5281/zenodo.7767202.
Table 2. Remotely sensed, in situ, and reanalysis data sets used in this paper.



| Data set | Link | Citation |
|---|---|---|
| Geostrophic current | https://data.marine.copernicus.eu/product/SEALEVEL_GLO_PHY_L4_MY_008_047/ | Pujol et al. (2016) |
| Ocean Waves Reanalysis data | https://data.marine.copernicus.eu/product/GLOBAL_MULTIYEAR_WAV_001_032/ | Law-Chune et al. (2021) |
| WindSat products | https://www.remss.com/missions/windsat/ | Wentz et al. (2013) |
| Quikscat products | https://www.remss.com/missions/qscat/ | Wentz et al. (2011) |
| Global Drifter data | https://www.aoml.noaa.gov/phod/gdp/hourly_data.php | Elipot et al. (2022) |
| OSCAR current | https://search.earthdata.nasa.gov/downloads/6873202440 | Johnson et al. (2007) |
| GlobCurrent product | https://tds0.ifremer.fr/thredds/GLOBCURRENT/GLOBCURRENT.html?dataset=GLOBCURRENT-L4-CUREUL_15M-ALT_SUM-V03.0 | Rio et al. (2014) |


**6 Concluding remarks**

It is understood that there are multi-scale dynamic processes in ocean surface current, among which large-scale geostrophic circulation and wind-induced Ekman flow are widely considered, while small-scale wave phenomena are ignored. In view of the facts, a new reconstruction product of the global ocean flow field at a depth of 15 m integrating Geostrophic, Ekman, Stokes, and Tidal components is produced in this research, leading to the following conclusions:

(1) Multi-scale dynamics-based GEST product of ocean surface current is reliable because of the



377 declining RMSE of the 0.25° resolution product which is found to be about 3.6 cm/s lower

378 than the OSCAR product, and 0.3 cm/s lower than the GlobCurrent. In particular, the inclusion

379 of wave-induced tidal currents and Stokes drift improves the precision of the reconstruction

380 mainly in equatorial and coastal regions.

381 (2) The verification of the wind-driven friction depth with local applicability in reconstructing

382 GEST product is significantly necessary, as evidenced by an 18 % increase in the correlation

383 coefficient between Ekman currents and drifter velocities.

384 (3) It has been revealed that high-resolution product with a 0.25°×0.25° cell has a closer agreement

385 to drifting buoys by revealing most of the mesoscale eddy kinetic energy missed by 1°

386 resolution product and is likely to be a better indicator of the real ocean circulation given the

387 improved accuracy of 5.62 cm/s.

388 It is worth pointing out that, in contrast to the existing ocean current estimation products mentioned

389 above, better results of the proposed GEST dataset at low to mid-latitudes are obtained

390 independently without assimilation of any drifter observation. In the future, complex mechanisms

391 (e.g., planetary waves, eddies, etc.) will be taken into account in the equatorial and strong current

392 regions to achieve a more accurate reconstruction of the global flow field.

**Author contribution**

394 GYW drafted the manuscript, developed the data processing, and designed the experiment. GC

395 contributed to conceptualizing the project, supervising, reviewing and editing the paper. CCC

396 developed the data processing and edited the paper. XYC edited and reviewed the paper. BXH

397 supervised and reviewed the paper.

**Competing interests**

399 The authors declare that they have no conflict of interest.

**Acknowledgments**

401 The authors acknowledge that WindSat data are produced by Remote Sensing Systems and

402 sponsored by the NASA Earth Science MEaSUREs DISCOVER Project and the NASA Earth



Science Physical Oceanography Program. RSS WindSat data are available at www.remss.com.
QuikScat (or SeaWinds) data are produced by Remote Sensing Systems and sponsored by the
NASA Ocean Vector Winds Science Team. Data are available at www.remss.com.
This paper also acknowledges funding from the National Natural Science Foundation of China
(Grant No. 42030406) and the International Research Center of Big Data for Sustainable
Development Goals (No. CBAS2022GSP01).

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
