# Peer review of "GEST: A multi-scale dynamics-based reconstruction of global ocean surface current"

_Earth System Science Data, 2023_

## Author Comment (AC1)

**Reply to referee #1 of**

**"GEST: A multi-scale dynamics-based reconstruction of global ocean surface current"**

**Authors:** Guiyu Wang et al.

June 19, 2023

(RC: referee comments | AC: authors comments)

**RC:** This paper presents a methodology to reconstruct ocean surface currents optimally combining different components of the ocean surface circulation : geostrophic, Ekman, tidal and wave-induced currents. Scientifically this work is sound, as it aims at providing a global surface currents product which can account locally for significant processes which could be missed relying only on a single/few of the aforementioned surface circulation components. I therefore recommend this paper for publication after considering the following major and minor issues.

**AC**: We highly appreciate your valuable comments and professional advice. Taking into consideration your suggestions and requests, we have made a substantial revision. Please find the specific details outlined below.

**Responses to General Comments:**

**RC:** Firstly, the metrics for evaluating the goodness of the data are mostly based on direct comparison with in-situ measured currents (drifting buoys), while the authors, already at the abstract level, mention the importance of both "high precision" and "fine resolution". I think inserting additional analyses (e.g. spectral analyses based on Fast-Fourier-Transform) to evaluate the effective resolution of the GEST data set could strengthen the manuscript (at least for the analyses presented in section 4.2);

**AC:** We added the results of the FFT-based spectral analyses to illustrate the effective resolving capabilities of our GEST data set. The results indicate a dominate signal scale of approximately 250 km and an equal minimum spatial scales of 10-50 km for GEST and GlobCurrent products under the same spatial resolution, which are both significantly better than the minimum spatial scales of 100-600 km for OSCAR product with 1° resolution. Specifically, figure A shows the wavenumber-power spectral density within the North Pacific Ocean (20°N-30°N,160°E-170°W) and the North Atlantic Ocean (30°N-50°N, 50°W-30°W). Figure B and figure C exhibit the latitudinal distribution of maximum and peak spectral wavenumbers of GlobCurrent (green dotted line), OSCAR current (blue dotted line), and GEST current (red dotted line) across each 5° band. The

maximum wavenumber indicates the smallest spatial scale at which the product can be resolved, while the peak spectral wavenumber represents the dominant signal scale within the product. Besides the capacity for spatial resolution, GEST current also shows lower reconstruction errors through accuracy verification.

[Figure]

Figure A: The GEST wavenumber-power spectral density (PSD) within the (a) North Pacific Ocean (20°N-30°N,160°E-170°W), and (b) the North Atlantic Ocean (30°N-50°N, 50°W-30°W).

[Figure]

Figure B: The maximum wavenumbers of GlobCurrent (green dotted line), OSCAR current (blue dotted line), and GEST current (red dotted line), by latitude bands.

[Figure]

Figure C: The zonal peak spectral wavenumbers for GlobCurrent (green dotted line), OSCAR current (blue dotted line), and GEST current (red dotted line), by latitude bands.

**RC:** While reading the results, I was concerned by the statistical significance of some of them. As an example, I may provide the specific case reported in figure 14 a. The RMSEs of the different datasets under evaluation are often few tenths of cm/s apart from each other. I was wondering if it is possible to add an information on the confidence level of the different RMSE computation (e.g. via bootstrap analysis). I think it could help readers understand when the GEST RMSE value is confirming the higher accuracy compared to other available surface currents datasets:

[Figure]

Figure D: The absolute error of OSCAR current (blue line), the GlobCurrent (green line), the GEST current (red line), by latitude bands. The shaded red, green, and blue areas indicates the 95% confidence interval for the absolute error.

AC: In accordance with your suggestion, we performed the bootstrap analysis to calculate the absolute error within the 95% confidence interval, represented by the color lines with shading in the above figure D. It is evident that the accuracy of the three products remains consistent within the equatorward regions of 20°S-15°N. Our GEST product has the highest accuracy in latitudes ranging from 20° to 50° in southern hemisphere and 10° to 40° in northern hemisphere by compared to the other two products. Additionally, its accuracy in higher latitude regions is comparable to that of GlobCurrent. Notably, the OSCAR current product exhibited the least accurate performance across nearly all latitudes.

RC: I struggled a bit to understand why the Authors wish to provide a data set with 1°x1° degrees spatial resolution. It was not that clear to me for which purposes/applications it was meant for. Could the Authors please explain?

AC: Among the four flow fields (geostrophic, Ekman, tidal currents, and Stokes drift) involved in the reconstruction, Ekman current has the lowest 1° spatial resolution after the local applicability analysis, and we initially attempted to reconstruct the flow field with its spatial resolution. However, the results are not very satisfactory. This 1°resolution data set can be considered as an interim product in the flow field reconstruction process, and also a contrast example of the effect of resolution on flow field reconstruction.

**Responses to Specific Comments:**

RC: I would recommend to further clarify section 3. I honestly struggled a bit to understand that section 3 presents the building blocks of the GEST product. Inserting few lines explaining this concept would be beneficial for the paper;

AC: We have added a further explanation for section 3 in the revised manuscript as below (line 142-147).

Revised: "Before ocean current reconstruction, four flow fields were temporally and spatially matched to drifter observations, and a series of data preprocessing and analyses were carried out. Then, the global correlation distribution of different flow fields with drifters was calculated. After that, the global reconstruction sub-models based on different ocean current combinations were constructed and validated, separately, to choose the best performing model in each 3° grid, which is finally used to reconstruct the sea surface current."

RC: I think the introduction is lacking part of the effort that has been done to reconstruct surface currents from tracers observations. I thus encourage the authors to consider few additional literature items (e.g. Bowen et al. 2002, Gonzalez-Haro et al. 2014, Liu et al. 2017, Rio and Santoleri 2018 and

references therein) and insert few lines in the introduction;

**AC:** These papers have been added to the reference list of our manuscript, and the description in the introduction section is as follows (line 50-54).

Revised: "...In addition, high-resolution Sea Surface Temperature (SST) products and Ocean Color (OC) images have also contributed to improving the accuracy of reconstructed ocean currents with methodologies from Maximum Cross Correlation technique and surface quasi-geostrophic theory. (Bowen et al., 2002; González-Haro andIsern-Fontanet, 2014;   Liu et al., 2017; Rio and Santoleri, 2018). ..."

**RC:** In the manuscript, there are some images (see e.g. figs. 1,3,5) where you claim you are presenting information on 1°x1° boxes. It seems to me the information has somehow been further smoothed spatially. Could the Authors please clarify the reasons behind this choice or, at least, clarify that directly in the text?

**AC:** These diagrams were automatically interpolated before, and they have been updated as below.

[Figure]

Figure 3: (a) Global mean distribution of the friction depth per 1° × 1°. (b) Proportion of friction depth up to 15 m in drifter observations.

[Figure]

Figure 1: Spatial distribution for the drifter observations per 1° × 1° from 1999 to 2019.

[Figure]

Figure 5: The correlation between Ekman currents and drifter observations per 3° grid (a) before and (b) after depth validation.

**RC:** A doubt is still related to figure 14 a/c/d: on average, it seems that the overall RMSE of GEST and Globcurrent datasets are equivalent in the latitudinal bands 30-50N and 40-60S, In particular, it seems that GEST and Globcurrent products show alternating better performances in the 30-50N band and equivalent performances in 40-60S band. Such areas are dominated by major current systems, thus relevant for assessing the quality of a surface current product. Could the authors please try to explain further such behaviour? I think it would be useful for users, in order to understand which data set should be used for a specific study area/application;

**AC:** We think the phenomenon that the RMSE of the GEST and Globcurrent products show alternating better performances in the region you mentioned may be related to the smaller amounts of drifter observations at middle latitudes that can be seen in figure 3 (b), leading to a lack of model reconstruction accuracy. Furthermore, the GlobCurrent data set assimilates drifter observations in the calculation of Ekman current, and we guess that it may results in an improved reconstruction accuracy in the westerly zone (30° N-50° N and 40° S-60° S) where Ekman current has high correlation with drifter observations.

**RC:** Line 152-153: this sentence is unclear to me. Which data set does not constitute a full independent validation for your reference field? Which reference field are the Authors referring to?

**AC:** We mean that previous studies have assimilated drifter observations to derive the ocean current product, while we have not used this method and have achieved similar accuracy, thus demonstrating the independence and strength of our algorithm.

**RC:** Line 159: What do you mean exactly with " Ekman layer reaches the position of the drogued drifters position of the drogue drifters"? Are you referring to the vertical position of the drogue? If so, please specify;

**AC:** Yes, "the position of the drogued drifters" refers to the vertical position of the drogue (i.e. 15m). We have specified it in the revised manuscript (line 184-185).

Revised: "A verification is necessary that the mixing depth of the Ekman layer reaches the vertical position of the drogue (i.e. 15 m)."

**RC:** Line 186: although one might guess what the Authors mean by "ocean components" I don't think it is appropriate to mention that one can compare "ocean components and drifter observations. I'd ask the Authors to rephrase the sentence;

**AC:** The corrections made are as follows (line 194-196).

Revised: " With the deepening of Ekman depth $h_e$, the correlation between ocean current velocities with different scales and drifter velocities shows a trend of increase and then decrease, ..."

**RC:** Figure 6: The legend should clearly mention what the Authors mean by the blue and red curves. Also, it seems the "Attenuation of experience" and theoretical formula are not precisely mentioned in the manuscript. This makes It hard to intercompare the figure with the findings reported in the text. Could the Authors please further clarify?

**AC:** The "empirical percentage method" and the "theoretical method" are further specified, as follows (line 205-217). In addition, we have also modified figure 6: Taking as an example in figure 6 (a), the Y-axis represents the correlation between geostrophic+ Ekman+Stokes zonal velocity and drifter zonal velocity, while the X-axis are the decay scale of Stokes drift using different decay methods (red curve: theoretical decay method (upper X-axis); blue curve: empirical decay method (lower X-axis)).

[Figure]

Figure 6: The zonal (top row) and meridional (bottom row) correlation of vector combinations of (a)-(b) geostrophic, Ekman currents, and Stokes drift, and (c)-(d) geostrophic, Ekman, and tidal currents, by theoretical decay method (red line) and empirical decay method (blue line). The upper x-axis represents decay depth Z of the theoretical decay method for Stokes drift, and the lower represents the percentage decay scale C of the empirical decay method for Stokes drift and tides.

Revised: "The reanalysis Stokes drift and tidal currents $u_0$ cover 0 m vertically and need to be attenuated, respectively, by the empirical decay method given in Eq. (4a), which is related to a decay scale c,

$$u = u_0 \ (1 - c\%), \tag{4a}$$

$$u_s = u_{s0} \exp \ (2kz). \tag{4b}$$

Additionally, there is a theoretical decay equation for Stokes drift that can be expressed in Eq. (4b), where $u_{s0}$ denotes the surface Stokes drift, $z$ is the profile derived from a monochromatic wave with

wavenumber $k$ and wavelength $\lambda=2\pi/k$ (Kukulka and Harcourt, 2017)."

RC: Line 203: reading the text it is not simple to understand what the Authors mean by "theoretical formula". Is this equation 4? Please specify and, in addition, I would provide more details for the "empirical percentage method", in order to help readers not familiar with that;

AC: We have corrected the descriptions, please check the responses to the above comment.

RC: Line 224: "Geostrophic currents act as the primary mechanism that form the ocean surface current field". I do not think it is appropriate to mention that the Geostrophic Currents are a mechanism that generate the ocean current field. I would rather say that they are a component of the total marine currents field;

AC: We have revised the sentence as below (line: 242-244).

Revised: "Geostrophic current is a major component of the ocean surface current field, and the Pearson correlation coefficient can reach nearly 0.98 in the regions with strong and persistent currents along the western boundary."

RC: Line 254: recalling what I mentioned in the "major comments" section. I would ask if the Authors could check whether this 0.3 cm/s RMS difference is significant or not;

AC: Following your suggestion we have added bootstrap analysis, which has been further described in the "general comments" section. In fact, the average observed velocity of the flow field is about 12.28 cm/s and the mean RMSE is approximately 2 orders of magnitude smaller than the true value, so the mean RMSE of 0.3cm/s is a small error. However, it is worth pointing out that we do not assimilate drifter observations in the reconstruction of ocean current, so the results are more independent and scientific.

RC: Table 1: I would ask the Authors to apply a minor change to the Table: please add the name of each sub-region, in order to help the readers locating the different sub-regions in a global map;

AC: Corrected. The updated table is as follows.

**Table 1.** Verified RMSE (cm/s) based on Sub-GE/ Sub-GES/ Sub-GET/ Sub-GEST models

| Reconstruction Model | Longitude and Latitude | | | |
|---|---|---|---|---|
| | 55° E-70° E 10° S-5° N | 80° E-100° E 45° S-60° S | 103° E-113° E 3° N-19° N | 125° E-142° E 22° N-37° N |
| | Southwest Maldives | Southwest Australia | Eastern Malaysia | Southern Kyushu Island |
| Sub-GE | 22.0333 | 9.1091 | 12.3418 | 11.7525 |

| | | | | |
|---|---|---|---|---|
| Sub-GES | 20.9319 | 9.2731 | 11.9773 | 11.9136 |
| Sub-GET | 23.3724 | 9.2368 | 12.4657 | 11.3579 |
| Sub-GEST | 20.6216 | 9.3872 | 12.0061 | 11.4845 |

| Reconstruction Model | Longitude and Latitude | | | |
|---|---|---|---|---|
| | 156° E-173° E 10° S-10° N Southwest Marshall Islands | 175° E-195° E 40° S-55° S Southeast New Zealand | 228° E-240° E 9° S-21° S Western Peru | 260° E-290° E 18° N-28° N Gulf of Mexico |
| Sub-GE | 16.9963 | 7.1911 | 6.5527 | 7.9553 |
| Sub-GES | 16.4886 | 7.3135 | 6.5541 | 8.0693 |
| Sub-GET | 17.0196 | 7.2609 | 6.5225 | 7.6361 |
| Sub-GEST | 16.5005 | 7.3945 | 6.5184 | 7.8609 |

**RC:** Lines 266-271: would it make sense/be possible for the Authors to add a global map that emphasizes the choice of the different sub-models combinations used for the global reconstruction?

**AC:** A global distribution of the zonal optimal sub-models combinations in Spring can be seen in below figure E, and has been added in the revised manuscript in section 3.5.

[Figure]

Figure E: The global distribution of the zonal optimal sub-models combinations in Spring.

**RC:** Figure 11/Line 290: I sincerely struggled a bit in understanding the meaning of synthetic vector/optimal vector. Could the Authors please further explain or harmonize the nomenclature of the variables?

**AC:** The "synthetic vector" means the synthesis of zonal and meridional components (i.e. $\sqrt{u^2 + v^2}$), while "optimal vector synthesis" is a flow field reconstruction model, which can choose the optimal vector synthesis combination of flow fields. The latter is the same as the optimal regression model and aims to select the best combination of flow fields. We have revised the caption of figure 11.

[Figure]

Revised: "Figure 11: The RMSE of the 1° reconstructed field of (a) zonal vector u, (b) meridional vector v, and (c) synthetic vector $\sqrt{u^2 + v^2}$, with the optimal combination of the regression model, and the 0.25° reconstructed field of (d) zonal vector u, (e) meridional vector v, and (f) synthetic vector $\sqrt{u^2 + v^2}$, with the optimal combination of vector synthesis model."

**Responses to Technical Comments:**

**RC:** Line 78: Maybe I would say " and the Globcurrent project products";

**AC:** We have corrected it as below (line 81-83).

Revised: "This global daily product covers the period of 2013-2019, with a 0.25° spatial resolution, and is compared with the OSCAR and the GlobCurrent project products."

**RC:** Line 108-110 (and elsewhere if necessary): Please remove the acronym CMEMS , keep Copernicus Marine Service instead;

**AC:** Corrected.

**RC:** Line 201: "wavenumber k and wavelength ?"  It seems this sentence is unfinished. Could the Authors please double check?

**AC:** We have added a detailed description of the wavelength (line 215-217).

Revised: "Additionally, there is a theoretical decay equation for Stokes drift that can be expressed in Eq. (4b), where $us_0$ denotes the surface Stokes drift, $z$ is the profile derived from a monochromatic wave with wavenumber $k$ and wavelength $\lambda=2\pi/k$ (Kukulka and Harcourt, 2017)."

**RC:** Figure 9: I would add x and y axes labels as Longitude and Latitude while It is redundant to repeat RMSE (cm/s) for each of the four sub-figures;

**AC:** We have corrected it, and please see figure 9.

[Figure]

Figure 9: RMSE of (a)-(d) Sub-GE, (e)-(f) Sub-GES, and (g)-(h) Sub-GET models.

**RC:** "The OSCAR near-surface current with a grid size of 1°on a 5 days basis use the quasilinear, quasi-steady sea surface momentum equations and improve the equatorial algorithm by fitting…" should be modified as follows: "The OSCAR near-surface currents product, with a grid size of 1° on a 5 days basis, uses the quasi-linear, quasi-steady sea surface momentum equations and improves the equatorial algorithm by fitting…"

**AC:** It has been corrected as follows (line 323-325).

Revised: "The OSCAR near-surface current product, with a grid size of 1° on a 5 days basis, uses the quasilinear, quasi-steady sea surface momentum equations and improves the equatorial algorithm by fitting 12 orthogonal polynomials (Johnson et al., 2007)."

---

## Author Comment (AC2)

**Reply to referee #2 of**

**"GEST: A multi-scale dynamics-based reconstruction of global ocean surface current"**

**Authors:** Guiyu Wang et al.

August 9, 2023

**Responses to Referee Comments:**

(RC: referee comments | AC: authors comments)

**RC1:** This paper presents a sea surface current product at a depth of 15 meters called GEST (Geostrophic-Ekman-Stokes-Tides) and compares it with other surface current products. This work shows some meaningful and interesting results, but there are still critical flaws present. Therefore, I believe that this paper is not suitable for publication in the journal Earth System Science Data, which aims to present original and high-quality datasets.

**AC1**: Thanks for your valuable comment. It is a great honor that you consider it interesting and meaningful to our ocean current product at a depth of 15 m reconstructed from drifter observations and the characteristics of Ekman current. Although more researchers are focusing on high-precision ocean current reconstructions, existing studies mainly utilize surface geostrophic and Ekman currents to model the physical inversion of global or regional oceans, neglecting the contribution of mesoscale eddies, sub-mesoscale dynamics, and small-scale wave motion. In particular, previous studies have demonstrated the effects of tidal current and Stokes drift on ocean current dynamics (e.g., Constantin, A., 2006; Sheehan et al., 2017; Onink et al., 2019), and any reconstruction model of the ocean current that ignores tidal current and wave-induced Stokes drift would be incomplete.

In addition, most of the presently similar ocean current products do not considerate the relative position of the wind-driven friction depths (ranging from a few meters to several hundred meters) versus the drogue of drifters (i.e., 15m). As a result, shallower wind-driven current is blended into the reconstruction process, which reduces the correlation with drifter observations.

Furthermore, drift measurements are used to correct for geostrophic and Ekman currents in previous products (e.g., GlobCurrent data), which deprive an independent validation process. In contrast, the GEST product has obtained comparable or even better results at low and mid-latitudes without assimilating any drift observations.

The GEST product also incorporates multiscale features and unifies the vertical depth of the reconstructed dataset by the verification for local applicability of Ekman current and the attenuation for tides and Stokes drifts, which result in a substantial improvement on previous products.

As detailed below, substantial responses and revisions have been made following your comments and suggestions.

**RC2:** As pointed out by the authors, there are presently similar data products available, such as OSCAR, GEKCO, and the GlobCurrent project. Concerning data precision, GEST does not fundamentally surpass GlobCurrent in terms of enhancement (Figure 14), as the resolution of satellite data remains unaltered. While the inclusion of tidal currents and Stokes drift might enhance accuracy in specific regions (Figure 8), this, however, results in a time span limited to only 2013-2019, which is significantly shorter than the extensive coverage of the traditional dataset (from 1993). Given comparable data accuracy, the study has not presented us with well-founded reasons to exclusively use this product. Personally, I might still favor the use of GlobCurrent (http://globcurrent.ifremer.fr/) which spans over 20 years.

**AC2:** Despite the obvious success of presently similar data products that have advanced the ocean current reconstruction process, the phenomenon of mesoscale eddies, sub-mesoscale dynamics you suggested in question 5, and small-scale fluctuations existing in the ocean have been neglected. The GEST dataset incorporates four ocean flow fields with emphasis on wave-induced Stokes drift as well as the tidal current, and finally achieves higher accuracy over coastal and equatorial regions, which is also endorsed by you.

As you mentioned, GEST does not completely outperform GlobCurrent product on a global scale in Figure 14, but alternates better performances in the latitudinal bands 30° N-50° N and 40° S-60° S. This is related to the fact that the GlobCurrent dataset utilizes drifter observations in the calculation of the CNES-CLS13 MDT and Ekman current, and the region where Ekman current shows high correlation with drifter observations lies roughly in the westerly zone (30° N-50N and 40° S-60° S), contributing to the high reconstruction accuracy of this latitudinal bands. Instead, the reconstruction accuracy of the GEST product without the assimilation of drift observations is about 4 cm/s and 0.3 cm/s higher over the OSCAR product and the GlobCurrent dataset. This suggests that the GEST product not only provides independent data validation but also improves reconstruction accuracy compared to the GEKCO and GlobCurrent products that utilize drift data and use it as a validation set.

Besides, GEST dataset is also characterized by the consideration for the local applicability of the Ekman current, ensuring that the Ekman layer reaches the vertical position of the drogue (i.e. 15 m). It improves the correlation of the Ekman current with the drifter observations by about 15 %.

Regarding the data spanning, it is available to provide the 0.25° resolution GEST dataset before 2013 as a supplementary product if anyone needs it. In our algorithm for ocean current reconstruction, a

portion of the data is used for model training and the rest is used for the prediction of the flow field. Typically, the data involved in training is not included in the output product, so the previous GEST product has only provided current forecasts since 2013.

**RC3:** The comparison of different velocity combinations in Section 3.5 yields insightful outcomes, clarifying the roles (positive or negative) played by Stokes drift and tidal currents. So, I recommend considering the submission of this paper to an alternative journal, rather than ESSD. In terms of the dataset's intrinsic merit, this study does not yield a novel and compelling product. Additionally, I propose visualizing the results in Table 1 using a bar chart, as it would provide a more intuitive depiction.

**AC3:** The Earth System Science Data (ESSD) journal encourages submissions on original data or data collections that are of sufficient quality. The intrinsic value of the GEST dataset lies in the accurate and independent reconstruction process, including the verification of the wind-driven friction depth with local applicability, the independent validation without assimilating drifter data, and the innovative reconstruction of the sea surface flow field from a multi-scale perspective. As you suggested, the comparison of reconstruction error for different combinations of models in Section 3.5 can demonstrate the role of tidal current and Stokes drift in the upper ocean. In addition, the importance of multiscale reconstruction can be confirmed by the higher global reconstruction accuracy compared to GlobCurrent and OSCAR data.

Moreover, following your suggestion, we visualized the results of Table 1 using the bar chart in Figure A1 (page 27 line 474). In contrast to abstract tables, a bar chart makes the performance of the four models visible at a glance.

[Figure]

Figure A1: The reconstructed RMSE (cm/s) based on Sub-GE (blue) / Sub-GES (orange) / Sub-GET (green) / Sub-GEST(red) models in Southwest Maldives, Southwest Marshall Islands, Eastern Malaysia, Southern Kyushu Island, Southwest Australia, Gulf of Mexico, Southeast New Zealand, and Western Peru.

**RC4:** The authors compare products with 1-degree resolution and 0.25-degree resolution and indicate that the 0.25-degree product better approximates the real ocean currents, which is not a surprising result. Undoubtedly, a 1-degree product cannot capture mesoscale eddies, and its accuracy is certainly lower than that of the 0.25-degree product. This fact is widely recognized and should not constitute the primary conclusion of this paper.

**AC4:** As the reviewer suggested the increased resolution of the flow field captures more mesoscale eddies with spatial scales around 50-200 km, which certainly contributes to the improvement of accuracy. These eddies have small horizontal scales, which usually require a horizontal resolution of at least the order of 10 km for ocean circulation models to depict the basic features more effectively. Thus the coarse resolution of the 1° data set does not effectively capture the mesoscale information, resulting in lower accuracy than the 0.25° resolution data.

In addition, our ocean current reconstruction model also contributes to the accuracy of the dataset, allowing the multi-scale information to be well exploited and the wind-driven current to be more accurate. Through quantitative studies, we revealed that the reconstruction error was reduced from 14.61 cm/s at 1-degree resolution to 9.36 cm/s at 0.25° resolution under the influence of the improved resolution and the multi-scale reconstruction algorithm.

We have updated the expression of the conclusion as follows,

Revised (page 1 lines 23-27): "Furthermore, by quantitatively analyzing the reconstructed products with 1° and 0.25° resolution, we find an improvement in accuracy of about 5.6 cm/s due to the multi-scale algorithm and higher resolution that reveals more details of ocean currents especially mesoscale eddy energy associated with geostrophic currents."

Revised (page 26 lines 459-464): "The quantitative analyses demonstrate a notable enhancement in reconstruction accuracy of about 5.6 cm/s due to the increase in resolution (from 1-degree to 0.25-degree) and the multi-scale reconstruction algorithm."

**RC5:** L68-70. "In the actual ocean … can be broadly divided into large-scale ocean circulations, micro scale internal waves and storm surges." This comment appears to lack thorough consideration, as it does not encompass mesoscale and sub-mesoscale processes. While only few observations can directly reflect sub-mesoscale processes, the authors should not dismiss their significant roles in the upper ocean, especially considering the increasing awareness of sub-mesoscale processes in recent years (e.g., JC McWilliams 2016). Relevant discussions are indispensable in this paper. In my view, incorporating sub-mesoscale processes could potentially significantly enhance the data accuracy.

**AC5:** Thanks for your professional comment. We have conducted a comprehensive literature review,

a portion of which is presented below. Further, we have incorporated and corrected the relevant content related to the mesoscale and sub-mesoscale processes within the revised manuscript.

The mesoscale dynamics dominated by mesoscale eddy energies have a strong influence on upper ocean dynamical processes and have been extensively studied by many scholars previously. Chen et al., (2021, 2022) proposed an independent identification scheme that relates eddy surface signature to its interior property, compensating for the lack of spatial resolution of altimeter products that leads to the missing identification of eddies. They found that roughly 1/4 of additional floats are identified by the verified Argo-alone criteria as onboard eddies outside altimetrically derived ones and the observed divergence and dispersion of eddy propagations are inextricably linked to ocean currents, winds, and topographic effects.

Meanwhile, sub-mesoscale dynamical processes have only recently been recognized and received more attention than large-scale circulation, mesoscale eddies and small-scale fluctuations. The generation of the sub-mesoscale current relies on mesoscale eddies and strong flow fields, which provides a dynamical conduit for energy transfer towards microscale dissipation and diapycnal mixing, accumulating an essential part of the total ocean kinetic energy.

For example, the sea surface density field contains rich variability over sub-mesoscale O (0.1-10) km length scales (e.g., McWilliams, J.C., 2016) that often manifest as density fronts and filaments. Sub-mesoscale density fronts are pervasive on continental shelves in high-resolution coastal models, observed within 10 km from shore. Wu et al. (2020) identified coastal density fronts (which they categorized into alongshore and cross-shore-oriented fronts, with the mean front length reaching 6-8 km, and depth <30 m) under weak wind conditions, and further analyzed their dynamical processes. Generated on the periphery of eddies formed during the growth of barotropic and baroclinic instabilities, many fronts are curved, with a radius of curvature comparable to the radius of the eddy. Shakespeare et al., (2016) analyzed the qualitative impact of curvature on the behavior and stability of density fronts in the ocean. They found that the curvature could change the cross-front (radial) force balance from a geostrophic balance to a cyclogeostrophic balance (a three-way force balance where the pressure and Coriolis forces must combine to provide a net inward centripetal force), as well as modify the potential vorticity of the system and the along front (angular) force balance.

Gula et al., (2015) studied the generation process of sub-mesoscale topographic vortex in the context of island wakes in the Gulf Stream region based on high-resolution realistic simulation. It can be viewed as generic for boundary slope currents moving anticyclonically/cyclonically around a basin (meaning that the flow has the coast on its left/right in the Northern Hemisphere), generating strong positive/negative vorticity within the bottom boundary layer, separating over complex topography, and forming a street of sub-mesoscale vortices. This process highlights a mechanism by which the interaction of a balanced flow with a sloping topography transfers energy from the larger-scale incident flows to sub-mesoscale flows and provides a way toward loss of balance and energy dissipation. Whereas Srinivasan et al., (2019) focused on sub-mesoscale vortex filaments at midlatitudes, with small bulk Rossby number in the lee of topography. It was found that the

generation and separation of bottom boundary layers form tilted shear layers with high vorticity and vertical shear, and the horizontal shear instability of these tilted shear layers on the slope generates sub-mesoscale vortical filaments. They also traced the evolution of unstable vortex filaments and the eventual formation of long-lived sub-mesoscale coherent vortices from both horizontal and vertical directions, suggesting that these processes are as ubiquitous in the oceans as the slope currents that produce it. However, they are unaccounted for in standard oceanic models on a global scale, and their impact will need to be parameterized.

As opposed to mesoscale eddies, sub-mesoscale information, which is densely and widely distributed, has not been captured by the current altimeter observations. Xia et al., (2022) developed an identification of sub-mesoscale eddies using SAR images based on the machine learning method (CAE-Net), and achieved better results than other models. Yurovsky et al. (2022) identified the structure of sub-mesoscale eddies along the Black Sea coast based on UAVs and analyzed the dynamics of the vortices, as well as the interactions with the wave-induced current. They found that the vortices with a diameter of ~200-400 m and an orbital velocity of ~0.15-0.30 m/s have a surprisingly high vorticity (the Rossby number Ro~15), and may have significant impacts on vertical circulation, energy transfer between large-scale motions and small-scale turbulence, and suspended matter transport.

Having gained a more detailed understanding of the dynamical processes at the mesoscale and sub-mesoscale described above, we modify our previous imprecise expressions as follows.

Revised (page 3 lines 76-77): "In the actual ocean, however, the movement of upper oceans is the result of multiple environmental driving mechanisms, and can be broadly divided into large-scale ocean circulations, mesoscale eddies, sub-mesoscale dynamics, and small-scale internal waves and storm surges."

Revised (page lines 429-433): "Also, the generation of the sub-mesoscale dynamics (e.g., density fronts, vortical filaments, and eddy streets) relies on mesoscale eddies and strong flow fields, which contributes to the energy transfer to microscale dissipation and diapycnal mixing, accumulating a significant part of the total kinetic energy of the oceans (Williams, 2016; Shakespeare et al., 2016; Srinivasan et al., 2019; Xia et al., 2022; Yurovsky et al., 2022)."

Revised (page 27 lines 468-471): "In the future, complex mechanisms (e.g., planetary waves, mesoscale eddies, and sub-mesoscale processes) will be integrated into standard ocean models as well as parameterized for their effects to enhance the reconstruction accuracy of the sea surface current."

Honestly, the purpose of this paper is to investigate whether the incorporation of tidal current and Stokes drift contributes to the upper ocean currents reconstruction. In the future, such sub-mesoscale processes will be integrated into ocean reconstruction models, and accurately quantified on a global scale to improve the reconstruction accuracy of the sea surface currents, with the help of the deep

mining and nonlinear parsing capabilities of the artificial intelligence techniques.

**RC6:** The inclusion of explanatory files and variable descriptions is crucial for a well-rounded and usable dataset. I downloaded the data provided by the authors (https://doi.org/10.5281/zenodo.7767202) and performed a preliminary processing. The current state of the dataset, containing only a data matrix in the NC file, is clearly inadequate for effective utilization. I strongly recommend that the authors take the necessary steps to rectify this issue and provide the essential context and information that will significantly enhance the dataset's quality and utility. If the authors truly intends to create a valuable dataset, they should invest more time in the dataset itself. It is evident that they has not drawn upon established exemplary datasets as references, such as (https://data.marine.copernicus.eu/product/MULTIOBS_GLO_PHY_REP_015_004/description).

**AC6:** Sincerely thank you for the reminder. There is a pivotal role for the data description file in the creation of datasets and it provides users with the ability to quickly understand the data set and utilize it efficiently.

We referred to the description of the GlobCurrent data and other products provided by the Copernicus Marine Service websites, finding that the websites provide detailed descriptions of variables such as geographical coverage, spatial scales, temporal extent, temporal resolution that we have neglected before. The explanatory documentation we added is as follows. We have also learned from the expression of the data in the NC file and updated the variable descriptions for the GEST product (https://doi.org/10.5281/zenodo.8262564).

Below is a basic descriptive information of the GEST data and we will follow up with further refinements.

**The Description of GEST Ocean Surface Current product**

**1. Summary**

The GEST Ocean Surface Current product represents an estimation of ocean surface current at 15 m depth, incorporating multi-scale physical processes such as Geostrophic and Ekman currents, wave-induced Stokes drift, and Tidal currents. This global daily product covers the period of 2013-2019, with a spatial resolution of 0.25°. It is likely to be a good indicator of the real ocean circulation.

**2. General information of the product specification**

| | |
|---|---|
| Product name | GEST Ocean Surface Current (G: Geostrophic current; E: Ekman current; S: Stokes drift; T: Tidal current) |
| Geographical coverage | Global Ocean
Lat 89.875° S to 89.875° N    Lon 0.125° E to 359.875° E |
| Spatial resolution | $0.25° \times 0.25°$ |
| Temporal extent | 1 Jan 2013 to 31 Dec 2019 |
| Temporal resolution | Daily |
| Depth | 15 m |
| Variables | Eastward sea water velocity (U)
Northward sea water velocity (V) |
| Feature type | Grid |
| Data assimilation | None |

| | |
|---|---|
| Format | NetCDF-4 |

**3. Details of variable descriptions**

| Variables name in the NetCDF file and Unit | Details |
|---|---|
| u_15m
(Centimetre per second) | Geostrophic velocity + Ekman velocity + Stokes velocity + Tidal velocity: zonal component
standard_name : eastward_sea_water_velocity
Land_mask=-9999
Fill_value=0 |
| v_15m
(Centimetre per second) | Geostrophic velocity + Ekman velocity + Stokes velocity + Tidal velocity: meridional component
standard_name : northward_sea_water_velocity
Land_mask=-9999
Fill_value=0 |
| Latitude | 89.875° S to 89.875° N |
| Longitude | 0.125° E to 359.875° E |

**References:**

Constantin, A. (2006). The trajectories of particles in Stokes waves. Invent. math. 166, 52-535, https://doi.org/10.1007/s00222-006-0002-5

Sheehan, P. M. F., Berx, B., Gallego, A., Hall, R. A., Heywood, K. J., Hughes, S. L., and Queste, B. Y.(2018). Shelf sea tidal currents and mixing fronts determined from ocean glider observations, Ocean Sci., 14, 225–236.

Onink, V., Wichmann, D., Delandmeter, P., & van Sebille, E. (2019). The Role of Ekman Currents, Geostrophy, and Stokes Drift in the Accumulation of Floating Microplastic. Journal of

Geophysical Research. Oceans, 124, 1474 - 1490.

Chen, G., Chen, X., & Huang, B. (2021). Independent eddy identification with profiling Argo as calibrated by altimetry. Journal of Geophysical Research: Oceans, 126, e2020JC016729. https://doi.org/10.1029/2020JC016729

Chen, G., X. Chen, and Cao, C. (2022). Divergence and Dispersion of Global Eddy Propagation from Satellite Altimetry. J. Phys. Oceanogr., 52, 705–722, https://doi.org/10.1175/JPO-D-21-0122.1

McWilliams, J. C., (2016). Submesoscale currents in the oceanProc. R. Soc. A.47220160117201 60117, http://doi.org/10.1098/rspa.2016.0117

Wu, X., Feddersen, F., and Giddings. S. N., (2021). Characteristics and Dynamics of Density Fr onts over the Inner to Midshelf under Weak Wind Conditions. J. Phys. Oceanogr., 51, 789 –808, https://doi.org/10.1175/JPO-D-20-0162.1

Shakespeare, C. J. (2016). Curved Density Fronts: Cyclogeostrophic Adjustment and Frontogene sis. J. Phys. Oceanogr., 46, 3193–3207, https://doi.org/10.1175/JPO-D-16-0137.1

Gula, J., Molemaker,M. J., and McWilliams, J. C. (2015), Topographic vorticity generation, sub mesoscale instability, and vortex street formation in the Gulf Stream, Geophys. Res. Lett., 42, 4054–4062, https://doi.org/10.1002/2015GL063731

Srinivasan, K., McWilliams, J. C., Molemaker, M. J., and Barkan, R. (2019). Submesoscale Vor tical Wakes in the Lee of Topography. J. Phys. Oceanogr., 49, 1949–1971, https://doi.org/1 0.1175/JPO-D-18-0042.1

Xia, L., Chen, G., Chen, X., Ge, L., and Huang, B. (2022). Submesoscale oceanic eddy detectio n in SAR images using context and edge association network. Front. Mar. Sci. 9:1023624. https://doi.org/10.3389/fmars.2022.1023624

Yurovsky, Y.Y., Kubryakov, A.A., Plotnikov, E.V., Lishaev, P.N. (2022). Submesoscale Current s from UAV: An Experiment over Small-Scale Eddies in the Coastal Black Sea. Remote S ens, 14, 3364. https://doi.org/10.3390/rs14143364

---

## Author Comment (AC3)

**Reply to Editor of**

**"GEST: A multi-scale dynamics-based reconstruction of global ocean surface current"**

Authors: Guiyu Wang et al.

September 9, 2023

**Responses to Editor Comments:**

**(EC: Editor Comments | AC: Authors Comments)**

**EC:** In addition to what has already been written by the two referees (with whom I fully agree), I would like to point out that the power spectrums shown by the authors leave me very perplexed. One can clearly see that very few points were used for their calculation (a spatial resolution of 0.25 deg means only 40 points over a distance of 10 degrees) and that the interpretation of the resulting effective resolution is wrong for the following reason:

The effective resolution of a spatial or temporal series is not the last point plotted in the PS D figures (the Nyquist frequency I suppose i.e., two times the grid resolution of about 50 k m), but the frequency at which the spectrum slope drops down. In this case this is not clear given the few points used to calculate the psd. Perhaps a vague indication of a change of slope could be at wavelengths of about 100 km, but certainly the spectrum shown does not allow this to be stated with certainty. My suggestion is to compute the power spectrum in a n area of the ocean where the longest series can be produced, compute a spectrum for each single date and finally average all the spectra. As an example, see figure 2 of Yang et al. "Sea Surface Temperature Intercomparison in the Framework of the Copernicus Climate Ch ange Service (C3S)" available at https://journals.ametsoc.org/view/journals/clim/34/13/JCLI-D-20-0793.1.xml.

In addition, the authors should better explain the concept of "training" mentioned in the response to referee 2 not described in the paper. In this regard, I also suggest publishing the entire time series mentioning the period used for the least-squares linear regression.

Considering that, in its current form, this article is between 'major revision' and 'rejected', I suggest that the authors, if they still intend to resubmit a revised version of the article, consider all referee comments very carefully in order to produce a new paper that fully satisfies them.

**AC1**: Thanks for your valuable comments. Following your suggestion, we have expanded the spatial and temporal scale of the dataset used to compute the power spectrum and averaged the spectra for each single date. By referring to Yang et al. (2021), the wavenumber-power spectral density curve has been modified and the relevant description has been revised in the response to referee 1.

Figure A3: The GEST wavenumber-power spectral density (PSD) within the (a) the North Atlantic region ( $60^{\circ}$  N- $70^{\circ}$  N,  $90^{\circ}$  W- $0^{\circ}$  W), and (b) the North Indian Ocean and Pacific Ocean ( $10^{\circ}$  N- $20^{\circ}$  N,  $30^{\circ}$  E- $80^{\circ}$  W). The black solid line represents the theoretical Surface Quasi Geostrophic (SQG) approximation spectra (the k-5/3 power law).

Figure A3 (a) shows the averaged wavenumber-spectral density for the North Atlantic region (60° N-70° N, 90° W-0° W, about 40×360 points per day) for the year 2013. The zonal component of the GEST (red solid line) and GlobCurrent data (blue solid line) in this region starts separating from the theoretical Surface Quasi Geostrophic (SQG) approximation spectra (the k-5/3 black solid line) at a wavelength scale of ~50 km. Whereas the meridional component of the GEST product (red dashed line) separates from the SQG approximation spectrum at a much smaller wavelength scale and starts to flat (~35 km), indicating that the signal contains only noise and has no physical significance at the wavelength shorter than 35km. Similarly, Figure A3 (b) provides the power spectrum located in the North Indian Ocean and Pacific Ocean (10° N-20° N, 30° E-80° W, about 40 × 1000 points per day). The power spectral density stays closer to the -5/3 slope at wavelengths around 100 km until the gradient becomes flat around 62 km. Note that the GEST and GlobCurrent products present mostly similar energy spectra because of the same spatial resolution.

In addition, the concept of "training" mentioned in the response to referee 2 refers to the process based on least-squares linear regression. We have published the corresponding dataset used during the least squares fitting process (https://doi.org/10.5281/zenodo.8329991) and updated the relevant expressions in the final response to referee 2.

Moreover, we attempted to expand the time series for the regression and updated the reconstruction algorithm (an example of the new model distribution is shown in Figure 10). The new latitudinal RMSE and correlation are shown in Figure 14 (a) and Figure 14 (c)-(d). It can be seen that the accuracy has a better improvement in the southern hemisphere and at high latitudes in the northern hemisphere.